# The Current Status of Adjuvant Chemotherapy for Colorectal Cancer in Japan: A Paradigm Shift from Oral Fluoropyridine Single Therapy to the Oxaliplatin Regimen

**DOI:** 10.3390/cancers17030518

**Published:** 2025-02-04

**Authors:** Nobuhisa Teranishi, Hiroyuki Uetake

**Affiliations:** Department of Clinical Research, National Hospital Organization (NHO), Disaster Medical Center, 3256 Midori-cho, Tachikawa 190-0014, Japan; n-tera@k4.dion.ne.jp

**Keywords:** colorectal cancer, adjuvant therapy in Japan, L-OHP, 5-FU + leucovorin, ACHIEVE trial

## Abstract

The effectiveness of oxaliplatin (L-OHP) has been reported overseas; however, in Japan, the prognosis of colorectal cancer (CRC) patients is reported to be good, and there has been a long debate about the applicability of L-OHP combination therapy in Japan. In recent years, the results of the ACHIEVE trial have become clear, and the standard consensus in Japan establishes L-OHP combination therapy for a duration of 3 months as the adjuvant treatment for CRC.

In Japan, the following regimens are covered by insurance for postoperative adjuvant chemotherapy for colorectal cancer: 5-FU + leucovorin (LV), UFT + LV, capecitabine, FOLFOX, and CAPOX. The effectiveness of oxaliplatin (L-OHP) has been reported overseas; however, in Japan, the prognosis of colorectal cancer (CRC) patients is reported to be good, and there has been a long debate about the applicability of L-OHP combination therapy in Japan. However, in recent years, the results of the ACHIEVE trial [1], a large-scale clinical trial conducted in Japan, have become clear, and the standard consensus in Japan establishes CAPOX or FOLFOX for a duration of 3 months as the adjuvant treatment for CRC. This article outlines the changes in postoperative chemotherapy for CRC in Japan.

## 1. L-OHP Combination Therapy

Fluoropyridine (FU) preparations have long been the standard treatment for postoperative adjuvant chemotherapy for CRC; however, since 2000, studies have been conducted to determine whether L-OHP combination therapy has an additional effect in postoperative adjuvant chemotherapy for advanced CRC. The MOSAIC study [2] investigated the effect of adding L-OHP to 5-FU + leucovorin (LV) as a treatment method for patients with Stage II/III colon cancer, and the 5-year DFS was significantly better in the FOLFOX group (5-year DFS: FOLFOX group—73.3%; LV5FU2 group—67.4%; HR = 0.80). By stage, the FOLFOX group showed a significantly better DFS in Stage III patients (5-year DFS: FOLFOX group—66.4%; LV5FU2 group—58.9%; HR = 0.78). Overall, no significant difference was observed for Stage II patients; however, when limited to high-risk Stage II patients, a favorable trend was observed in the FOLFOX group (5-year DFS: FOLFOX group—82.3%; LV5FU2 group—74.6%; HR = 0.72). Regarding 6-year OS, no difference was observed between the two groups in Stage II; however, in Stage III, the FOLFOX group was significantly better (FOLFOX group—72.9%; LV5FU2 group—68.7%; HR = 0.80), and a significant difference was also observed overall (*p* = 0.046). From the results above, the additional effect of L-OHP was observed in Stage III cancer patients, and it may also be effective in high-risk cases of patients with Stage II cancer. However, it has been reported that neurotoxicity, an adverse event of L-OHP, persists in more than 15% of cases with Grade 1–3 peripheral neuropathy, even four years after the end of chemotherapy. Similarly, in the NSABP C-07 trial [3], FLOX therapy (which entails the administration of L-OHP every other week to the RPMI regimen) was compared with the RPMI regimen, and the overall 5-year DFS was 69.4% in the FLOX group and 64.2% in the 5-FU/LV group, showing an additional effect of 5.2%. In addition, in the NO16968 (XELOXA) study [4], the CAPOX group showed a 6.3% increase in 5-year DFS compared to the bolus 5-FU group (Mayo or RPMI regimen) (median follow-up period = 57 months; HR = 0.80), and a 7% increase in 7-year DFS (56% vs. 63%, *p* = 0.004). Regarding the long-term survival observations (10 years and 6 years, respectively) in the MOSAIC and NO16968 studies, the differences in DFS and OS in the FOLFOX and CAPOX groups became larger over time compared to the group treated with FU alone [5]. This suggests that L-OHP provides a definite effect in postoperative adjuvant chemotherapy for Stage III CRC. In addition, a comparative study was conducted in Japan between high-risk Stage III patients treated for 6 months with either SOX therapy or UFT + LV therapy; however, the superiority of SOX therapy was not demonstrated [6]. The reasons for this result may be related to the fact that the amount of oxaliplatin in SOX therapy was low, at 100 mg/m^2^, and that S-1 is not proven to be equivalent to capecitabine in CRC adjuvant chemotherapy [7].

## 2. Optimal Administration Duration

The IDEA collaboration was an integrated analysis of six phase III studies (SCOT, TOSCA, HORG, CALGB/SWOG 80702, IDEA France, and ACHIEVE) in Stage III colon cancer patients in 12 countries; it compared the administration of CAPOX or FOLFOX [8] for durations of both 3 and 6 months. Non-inferiority was not demonstrated for 3-year DFS (HR = 1.07), which was 74.6% for the 3-month group and 75.5% for the 6-month group. However, in the low-risk group with T1-3 and N1 disease, non-inferiority was demonstrated for the 3-month group (HR = 1.01; 3-year DFS (3-month group: 83.1%; 6-month group: 83.3%)). In subgroup analysis, treatment for 6 months was superior to treatment for 3 months (HR = 1.16); however, for the CAPOX treatment method, non-inferiority was demonstrated for a treatment duration of both 3 months and 6 months (HR = 0.95). In the final analysis, non-inferiority was not demonstrated for 5-year OS (HR = 1.02; 3-month group: 82.4%; 6-month group: 82.8%), but the absolute difference was 0.4%. It was found that the 5-year DFS (HR = 1.08) was similar to that of previous results. These results support the standard clinical use of CAPOX therapy for a duration of 3 months. In the ACHIEVE study conducted in Japan (treatment regimen: CAPOX 75%, mFOLFOX6 25%), no significant difference was observed in 5-year DFS (HR = 0.95; 3-month group: 75.2%; 6-month group: 74.2%) or 5-year OS (HR = 0.91; 3-month group: 87.0%; 6-month group: 86.4%) [1]. The occurrence of adverse events was significantly lower in patients with a shorter treatment period, and the incidence of a grade 2 or higher level of neurotoxicity was significantly lower in the 3-month group with FOLFOX and CAPOX (3 months vs. 6 months: FOLFOX 12.6% vs. 36%; CAPOX 14% vs. 36.2%). In addition, the incidence of diarrhea, neutropenia, thrombocytopenia, nausea, mucositis, and hand–foot syndrome was significantly lower in the 3-month group.

In Japan, the use of L-OHP combination therapy in the treatment of CRC has been debated for a long time due to the good surgical outcomes of CRC. However, in the ACTS-CC trial [9] (UFT + LV or S-1), in the T3N1 group, the 5-year DFS was 77.1% and the 5-year OS was 90.9% (which is about 75% of the low-risk group), while in the ACIEVE trial, the 5-year DFS was 88.8% and the 5-year OS was 94.1% (Table 1).

Table 2 shows the frequency of side effects of both CAPOX therapy (3 months) and UFT/LV or S-1. In any item, the difference between CAPOX therapy and UFT/LV or S-1 therapy is small. In addition, the frequency of Grade 3 or higher is also low for peripheral neuropathy and hand–foot syndrome, which are the unique side effects of CAPOX therapy.

In the ACTS-CC trial (UFT + LV or S-1), the 5-year DFS and OS of T3N1 (about 75% of the low-risk group) was about 77.1% and 90.9%, respectively, (Ref. [9]).

Therefore, the following consensus has been reached by Japanese experts: “The standard treatment for postoperative adjuvant chemotherapy of CRC in Japan is 3 months of CAPOX therapy (or FOLFOX therapy). For cases with a high risk of recurrence, up to 6 months administration is recommended. For lowest risk patients, patients with complications, and elderly people, the option of administering FU alone is acceptable”.

## 3. FU Alone (Oral Drugs)

The usefulness of oral FU preparations has been reported, and their use in postoperative adjuvant chemotherapy has also been promoted. The NSABP C-06 study and the JCOG0205 study conducted in Japan demonstrated the equivalence of UFT/LV to 5FU/LV, while the X-ACT study demonstrated the equivalence of capecitabine to 5FU/LV. In terms of oral medications, the ACTS-CC study demonstrated the non-inferiority of S-1 to UFT/LV [9]; however, the JCOG0910 study [7] did not demonstrate the non-inferiority of S-1 to capecitabine. Based on the above results, three regimens are available for use: UFT/LV therapy, capecitabine therapy, and S-1 therapy. The standard treatment for oral adjuvant chemotherapy is capecitabine, but the adverse events differ. For example, UFT/LV causes liver dysfunction; capecitabine causes hand–foot syndrome; and S-1 causes diarrhea, stomatitis, and a loss of appetite.

## 4. High-Risk Stage II Colon Cancer

The 5-year survival rate for Stage II colon cancer is relatively good at 88.2%, but recurrence occurs in 15% of patients. The usefulness of postoperative adjuvant chemotherapy has been examined in the QUASAR, SACURA, and MOSAIC studies [3], as well as in the NASABP C-07 study [4]; however, its usefulness was not demonstrated, and it is not recommended for use in a uniform manner. However, even among Stage II colon cancer patients, there are patients with a high risk of recurrence, and the ASCO, ESMO, and NCCN guidelines list risk factors such as the presence of fewer than 12 dissected lymph nodes, T4, poorly differentiated adenocarcinoma, signet ring cell carcinoma, mucinous carcinoma, perforation, vascular lymphatic invasion, paraneural invasion, positive margins, high CEA levels, and tumor budding. In the JFMC46-1201 study conducted in Japan for high-risk Stage II colon cancer, UFT/LV therapy was compared with surgery alone, with risk factors of T4, perforation/penetration, poorly differentiated adenocarcinoma, mucinous carcinoma, and fewer than 12 dissected lymph nodes. It was shown to significantly extend DFS, but no difference was observed in OS. In a multivariate analysis, postoperative adjuvant chemotherapy was an independent factor in extending OS and disease-free survival (DFS). Although the definition of “high risk” remains controversial, adjuvant chemotherapy is recommended for selected cases of Stage II colon cancer.

## 5. Summary and Future Prospects

As mentioned above, the use of L-OHP combination therapy in Japan has long been debated, but the standard treatment in Japan has been established as CAPOX or FOLFOX for 3 months based on data from various clinical trials.

The IDEA pooled analysis showed data suggesting a difference in efficacy between FOLFOX and CAPOX in adjuvant chemotherapy for colon cancer. Accordingly in the ESMO Early Colon Cancer Treatment Recommendations [11], the stated length of oxaliplatin-based adjuvant treatment for stage III colon cancer based on the IDEA data may be tailored to 3 months for CAPOX (T1-3 N1 disease), 6 months for CAPOX (T4 or N2 disease), or 6 months for FOLFOX (T4 or N2 disease), and ‘patients with high risk stage II colon cancer may be considered for 3 months of CAPOX, as the IDEA pooled analysis showed non-inferiority of 3 months of CAPOX and inferiority of 3 months of FOLFOX when compared with 6 months of FOLFOX, with all the limitations of post-hoc analyses’.

In the future, further optimization of postoperative adjuvant chemotherapy will likely be studied, taking into account postoperative circulating tumor DNA, etc.

## Figures and Tables

**Table 1 cancers-17-00518-t001:** Results of 5-year DFS and OS in the ACHIEVE trial compared to the ACTS-CC trial.

ACHIEVE Trial	Low Risk: T1-3 and N1; High Risk: T4 and/or N2								
	5-Year DFS					5-Year OS			
Duration of L-OHP Administration	3 Months		6 Months	HR		3 Months		6 Months	HR
All	75.2	vs.	74.2	0.95	All	87.0	vs.	86.4	0.91
mFOLFOX6	68.6	vs.	69.7	1.04	mFOLFOX6	83.2	vs.	84.6	0.99
CAPOX	77.4	vs.	75.8	0.91	CAPOX	88.3	vs.	87.0	0.87
Low risk	86.5	vs.	84.8	0.85	Low risk	92.7	vs.	91.8	0.86
mFOLFOX6	79.2	vs.	84.3	1.41	mFOLFOX6	88.3	vs.	89.5	1.26
CAPOX	88.8	vs.	85.0	0.70	CAPOX	94.1	vs.	92.5	0.71
High risk	60.7	vs.	61.5	1.04	High risk	79.8	vs.	79.8	0.96
mFOLFOX6	56.3	vs.	55.9	1.01	mFOLFOX6	77.5	vs.	79.9	0.91
CAPOX	62.4	vs.	63.7	1.07	CAPOX	80.8	vs.	79.8	0.99

**Table 2 cancers-17-00518-t002:** Incidence of adverse events (%).

	CAPOX (3 Month) [8]	UFT/LV [10]	S-1 [10]
Any	24.2	16.0	14.4
Diarrhea	7.4	5.5	4.4
Nausea	3.0	1.2	1.6
Vomiting	2.0	0.8	0.8
Thrombocytopenia	3.0	0.4	0.1
PSN *	2.6	Not listed	Not listed
HFS **	0.7	Not listed	Not listed

* PSN; peripheral sensory neuropathy ** HFS; hand–foot syndrome.

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
