# Peer review of "The Current Status of Adjuvant Chemotherapy for Colorectal Cancer in Japan: A Paradigm Shift from Oral Fluoropyridine Single Therapy to the Oxaliplatin Regimen"

_cancers, 2025, doi:10.3390/cancers17030518_

Round 1

Reviewer 1 Report

Comments and Suggestions for Authors

Dear Authors,

in my opinion this manuscript could be accepted for publication. However, although a statement made by Japanese experts is reported, in summary should be reported that CAPOX and FOLFOX are considered different for efficacy. For example, ESMO guidelines recommend 6 months FOLFOX and 3 or 6 months CAPOX for high risk stage II colon cancer and 6 months FOLFOX for low and high risk stage III while 3 months CAPOX for the low risk. 

Author Response

Comment; 

Dear Authors, in my opinion this manuscript could be accepted for publication. However, although a statement made by Japanese experts is reported, in summary should be reported that CAPOX and FOLFOX are considered different for efficacy. For example, ESMO guidelines recommend 6 months FOLFOX and 3 or 6 months CAPOX for high risk stage II colon cancer and 6 months FOLFOX for low and high risk stage III while 3 months CAPOX for the low risk.

Response

We deeply appreciate and agree with the reviewer's important comments. Following their suggestions, we added "The IDEA pooled analysis showed data suggesting a difference in efficacy between FOLFOX and CAPOX in adjuvant chemotherapy for colon cancer. Accordingly in the ESMO Early Colon Cancer Treatment Recommendations, it is stated 'the length of oxaliplatin-based adjuvant treatment for stage III colon cancer based on the IDEA data may be tailored to 3 months for CAPOX (T1-3 N1 disease), 6 months for CAPOX (T4 or N2 disease), or 6 months for FOLFOX (T4 or N2 disease), and ‘patients with high risk stage II colon cancer may be considered for 3 months of CAPOX, as the IDEA pooled analysis showed noninferiority of 3 months of CAPOX and inferiority of 3 months of FOLFOX when compared with 6 months of FOLFOX, with all the limitations of post-hoc analyses’ " to the Summary and future aspect section (in red), and added the ESMO guidelines to the references (No 11 in red).

Reviewer 2 Report

Comments and Suggestions for Authors

The Authors of this paper describe paradigm shift in adjuvant chemotherapy for colorectal cancer in Japan. They summarised the most important papers dealing with adjuvant chemotherapy including trial conducted in Japan. The oncological results are clearly summarised in Table 1 and adverse events in Table 2. Other important issue is chemotherapy in high risk stage II patients. Chemotherapy in this group of patients improved DFS but not OS in Japan. Moreover, the Authors clearly demonstrated why 3 months of adjuvant CAPOX or FOLFOX regimens are sufficient for optimal combined treatment of cancer with limited toxicity. In my opinion the article should be published as it describes current approach to combined treatment of colorectal cancer in Japan and major studies that proved effectiveness of adjuvant treatment.

Author Response

Comment

The Authors of this paper describe paradigm shift in adjuvant chemotherapy for colorectal cancer in Japan. They summarised the most important papers dealing with adjuvant chemotherapy including trial conducted in Japan. The oncological results are clearly summarised in Table 1 and adverse events in Table 2. Other important issue is chemotherapy in high risk stage II patients. Chemotherapy in this group of patients improved DFS but not OS in Japan. Moreover, the Authors clearly demonstrated why 3 months of adjuvant CAPOX or FOLFOX regimens are sufficient for optimal combined treatment of cancer with limited toxicity. In my opinion the article should be published as it describes current approach to combined treatment of colorectal cancer in Japan and major studies that proved effectiveness of adjuvant treatment.

Response

We would like to express our sincere gratitude to the reviewers for reviewing our paper and picking out the important points.